# Antibacterial Activity and Mode of Action of Lactoquinomycin A from *Streptomyces bacillaris*

**DOI:** 10.3390/md19010007

**Published:** 2020-12-24

**Authors:** Beomkoo Chung, Oh-Seok Kwon, Jongheon Shin, Ki-Bong Oh

**Affiliations:** 1Department of Agricultural Biotechnology, College of Agriculture and Life Sciences, Seoul National University, Seoul 08826, Korea; beomkoo01@snu.ac.kr; 2Natural Products Research Institute, College of Pharmacy, Seoul National University, Seoul 08826, Korea; ideally225@snu.ac.kr

**Keywords:** *Streptomyces bacillaris*, lactoquinomycins, methicillin-resistant *Staphylococcus aureus*, dual-reporter system, DNA intercalation

## Abstract

This study aims to isolate and identify the structure of antibacterial compounds having potent activity on methicillin-resistant *Staphylococcus aureus* (MRSA) from marine actinomycetes, and also to identify their mode of action. Lactoquinomycin A (LQM-A) (compound **1**) and its derivatives (**2**–**4**) were isolated from marine-derived *Streptomyces bacillaris* strain MBTC38, and their structures were determined using extensive spectroscopic methods. These compounds showed potent antibacterial activities against Gram-positive bacteria, with MIC values of 0.06–4 μg/mL. However, the tested compounds exhibited weak inhibitory activity against Gram-negative bacteria, although they were effective against *Salmonella enterica* (MIC = 0.03–1 μg/mL). LQM-A exhibited the most significant inhibitory activity against methicillin-resistant *Staphylococcus aureus* (MRSA) (MIC = 0.25–0.5 μg/mL), with a low incidence of resistance. An in vivo dual-reporter assay designed to distinguish between compounds that inhibit translation and those that induce DNA damage was employed to assess the mode of action of LQM-A. LQM-A-induced DNA damage and did not inhibit protein synthesis. The gel mobility shift assay showed that LQM-A switched plasmid DNA from the supercoiled to relaxed form in a time- and concentration-dependent manner. These data suggest that LQM-A intercalated into double-stranded DNA and damaged DNA repair.

## 1. Introduction

Antibiotics that are either cytostatic or cytotoxic to microorganisms allow the human immune system to combat the pathogens [1]. Such antibiotics often inhibit the synthesis of fundamental components such as deoxyribonucleic acid (DNA), ribonucleic acid (RNA), proteins, and the cell wall, which is essential for bacterial survival or act to eliminate membrane potential [2]. Undoubtedly, antibiotics have contributed to improving the health of humanity and advancing human civilization, which has ushered in the antibiotic era [3]. Numerous classes of antibiotics have been produced over the last several decades in response to increased demand; however, the excessive popularization of antibiotics has led to the emergence of resistant microorganisms [4]. To overcome this crisis, the development and comprehensive analysis of antibiotics are urgently needed.

Aromatic polyketides, representative substances of type II polyketides, have significant therapeutic properties [5]. Tetracycline (Tet) and anthracycline-type doxorubicin (Dox) are typical examples of aromatic polyketides with pharmacological applications [6]. Their mechanisms of action are markedly different despite their structural similarity. Tet binds the 30S and 50S subunits of the ribosome and inhibits protein synthesis by interfering with the attachment of aminoacyl-transfer ribonucleic acid (aminoacyl-tRNA) to the A-site on the ribosome [7]. By contrast, Dox intercalates between DNA base pairs and inhibits macromolecular biosynthesis processes, such as DNA replication and RNA transcription [8]. Both compounds have been clinically applied, and Dox has been used as one of the most effective anticancer drugs [9]. Lactoquinomycin A (LQM-A), a member of the pyranonaphthoquinone family synthesized via type II polyketide synthases, was first reported from *Streptomyces* and found to exhibit significant anticancer and antibacterial activities [10,11]. The structure of LQM-A was revised, indicating that D-angolosamine is attached at C-8 of the naphthoquinone ring, based on the two-dimensional ROESY experiment [12]. While derivatives of LQM-A have been continuously discovered, their mode of action has not been clearly characterized.

During our search for bioactive secondary metabolites from marine-derived actinomycetes, we identified four lactoquinomycins (**1**–**4**) from *Streptomyces bacillaris* strain MBTC38, which was collected from marine sediment from Jeju Island, Republic of Korea. It has been reported that *S. bacillaris* isolated from terrestrial and marine habitats produce structurally unique natural compounds with diverse biological activities [13,14,15]. *S. bacillaris* strain RAM25C4 was isolated from marine sediments for controlling methicillin-resistant *Staphylococcus aureus* (MRSA) and multidrug-resistant bacteria. In HPTLC (high-performance thin-layer chromatography) analysis, the presence of macrolides, terpenoids, and quinolones was identified in the ethyl acetate extract of RAM25C4 culture [14]. In addition, *S. bacillaris* strain SNB-019 isolated from Galveston Bay, Texas produced bafilomycin analogs that were responsible for the autophagy inhibitory activity [15]. However, to the best of our knowledge, lactoquinomycins are the first to be reported from *S. bacillaris*. These lactoquinomycins showed significant activities and a low incidence of resistant strains among Gram-positive bacteria including methicillin-resistant *Staphylococcus aureus* (MRSA). An in vivo dual-reporter assay indicated that LQM-A does not inhibit protein synthesis, but instead induces DNA damage, similar to Dox. Furthermore, LQM-A was found to intercalate DNA based on the ethidium bromide displacement assay. The intercalation effect of LQM-A changed the form of plasmid DNA from supercoiled to relaxed.

## 2. Results

### 2.1. Taxonomy of MBTC38

The 16S rDNA of strain MBTC38 was amplified using PCR and sequenced. After a BLAST sequence comparison, strain MBTC38 showed 100% identity to *Streptomyces bacillaris* strain NBRC13487 (type strain, GenBank accession number: NR041146.1) and ATCC15855 (type strain, GenBank accession number: CP029378.1). Thus, this strain was designated *S. bacillaris* strain MBTC38 (GenBank accession number: MK402083.1).

### 2.2. Isolation and Structural Elucidation of Compounds ***1**–**4***

*S. bacillaris* strain MBTC38 was cultured in colloidal chitin liquid medium at 28 °C for 7 days with shaking and fractionated twice with equal volumes of ethyl acetate. After evaporation of the solvent, the crude extract was separated using semi-preparative HPLC to yield four compounds. Based on combined spectroscopic analyses including NMR, ultraviolet (UV), and mass spectrometry, compounds **1**–**4** were identified as lactoquinomycin A (**1**) [11], lactoquinomycin B (**2**) [16], N-methyl lactoquinomycin A (**3**) [17], and menoxymycin A (**4**) [18] (Figure 1). The spectroscopic data for these compounds were in good agreement with those in the literature (Appendix A).

### 2.3. Antibacterial Activities of Compounds ***1**–**4***

The antimicrobial activities of compounds **1**–**4** were evaluated against pathogenic bacterial strains including MRSA. These compounds displayed significant antibacterial activities against the Gram-positive strains tested (*S. aureus*, *E. faecium*, and *E. faecalis*), with MIC values of 0.06–4 μg/mL (Table 1). However, the tested compounds exhibited weak to no inhibitory activity against Gram-negative bacteria such as *E. coli* (MIC = 16–≥ 32 μg/mL) and *K. pneumoniae* (MIC ≥ 32 μg/mL), although they were effective against *S. enterica* (MIC = 0.03–1 μg/mL) when ampicillin and tetracycline were used as positive controls. These compounds also showed significant inhibitory activities against all MSSA and MRSA strains (Table 2). In particular, LQM-A (**1**) had significant and broad antibiotic effects on MRSA, with MIC values of 0.25–0.5 μg/mL, indicating that it is more potent than important antibiotics such as daptomycin (MIC ≥ 32 μg/mL), platensimycin (MIC = 4–8 μg/mL), linezolid (MIC = 1–2 μg/mL), and ciprofloxacin (MIC = 0.25–>32 μg/mL).

### 2.4. Multi-Step Resistance Development and Time-Kill Kinetics

Resistance development profiles for compounds **1**–**4** against MRSA strain ATCC43300 were determined (Figure 2a). In the present study, resistance was defined as a > 4-fold increase in the initial MIC [19,20]. For compounds 1, 3, and 4, 4-fold increases in the MIC (from 0.25 to 1 μg/mL, from 0.5 to 2 μg/mL, and from 1 to 4 μg/mL, respectively) were observed, indicating that resistance against compounds 1, 3, and 4 did not develop over 30 passages. Conversely, for compound 2 and ciprofloxacin, 8- and 256-fold changes relative to the initial MIC values (compound 2 = 2 μg/mL; ciprofloxacin = 0.25 μg/mL), respectively, were observed during the passage experiment.

To confirm the time-dependent antibacterial effects of LQM-A (1), which showed the strongest antibacterial activity, a time-kill kinetics experiment was conducted from 0 to 24 h. The time-kill profile of LQM-A showed a reduction in the number of viable cells for the first 8 h followed by a gradual increase up to 24 h, indicating that the inhibitory effect of LQM-A is much greater than those of the control antibiotics, tetracycline, and ciprofloxacin, which reduced the number of viable cells for the first 4 h. Meanwhile, vancomycin showed a bactericidal effect of decreasing the total number of viable cells at 24 h (Figure 2b).

### 2.5. Mode of Action of LQM-A

To assess the mode of action of LQM-A (1), we employed an in vivo dual-reporter assay, which was designed to identify mechanisms of inducing DNA damage or inhibiting translation [21,22]. This system operates on the principle that compounds inducing DNA damage cause the SOS response and activate the sulA promoter by inactivating LexA, which leads to the expression of RFP (shown in green pseudo-color). Meanwhile, compounds that inhibit translation trigger ribosome stalling at a tryptophan operon leader sequence mutated with two alanines (trpL-2Ala) and Katushka2S (far-red fluorescent) protein is expressed, as shown in red. LQM-A, the representative translation inhibitors Tet and erythromycin (Ery), and the typical DNA replication inhibitor Dox were spotted onto an agar plate already containing a lawn of the reporter strain. In contrast to Tet and Ery, LQM-A led to the appearance of pseudo-green fluorescence similar to that of Dox, indicating that LQM-A induces DNA damage rather than inhibiting translation (Figure 3a).

The mechanisms of LQM-A (1) and γ-actinorhodin, which share their main tetracyclic structure, are related to bacterial membrane potential [23,24]. Thus, we confirmed whether LQM-A increased membrane permeabilization with the membrane-associated fluorescent dye DiSC_3_ (5). In the present study, however, no change was observed in LQM-A-treated cell membranes, whereas nisin, which is known to generate pores in the cell membrane, led to enhanced fluorescence intensity over time (Figure 3b).

### 2.6. DNA Intercalation Effects of LQM-A

Based on its structural similarity to Dox, we expect that LQM-A intercalates into DNA, and therefore we conducted an EtBr displacement assay. EtBr, a fluorescent DNA intercalator, tends to be inserted between base pairs of the DNA double helix. This assay uses the reduction of fluorescence caused by compounds intercalated in DNA due to displacement of EtBr [25]. LQM-A (1) reduced fluorescence intensity by 36% as the concentration increased, similar to Dox. By contrast, Cip, which is known to act as a gyrase inhibitor, scarcely affected fluorescence intensity (Figure 4a). Additionally, 20-mers of poly-AT and poly-GC DNA were prepared to determine whether LQM-A prefers AT- or GC-rich regions of DNA. The poly-GC and LQM-A system showed a fluorescence reduction of 33%, while fluorescence was reduced by 42% in the poly-AT and LQM-A system (Figure 4b). We also performed a gel mobility shift assay to ascertain the intercalation activity of LQM-A. The plasmid pUC 19 switched from the supercoiled form to the relaxed form as time progressed. The relaxed form of the plasmid performed better than the supercoiled form at 4× the MIC value in the concentration-dependent experiment (Figure 5).

## 3. Discussion

Polyketides are a large family of secondary metabolites containing modified carbonyl groups or their precursors [26]. They are generally classified into three groups, types I, II, and III, depending on the enzyme types involved. Among them, aromatic polyketides, which are type II polyketides, have been reported to exhibit fascinating biological activities [6]. Numerous research groups have focused on aromatic polyketides with pharmacological potential, as they have advantages for clinical application including structural diversity, a broad range of therapeutic effects, and potential for genetic engineering [27]. In this study, four lactoquinomycins, which are a type of aromatic polyketides, were isolated from marine-derived *S. bacillaris* strain MBTC38 through activity-guided separation procedures and identified as LQM-A, lactoquinomycin B, N-methyl lactoquinomycin A, and menoxymycin A. These compounds showed potent inhibitory activities against pathogenic bacteria. Among these substances, known as compounds **1**–**4**, LQM-A exhibited the most significant antibacterial activity against Gram-positive bacteria including MRSA (MIC = 0.06–2 μg/mL). We also observed that the 4a,10a-epoxide structure played an important role in reducing biological activity; therefore, lactoquinomycin B unlike showed weak antibacterial activity (from 1 μg/mL to > 32 μg/mL).

LQM-A have been isolated from various species of *Streptomyces* as it was initially discovered from a soil actinomycete *S. tanashiensis* [28,29,30]. Furthermore, LQM-A has been reported to exhibit antimicrobial activities against Gram-positive strains (S. aureus FDA209, Bacillus subtilis PCI219, Corynebacterium xerosis) with MIC values of 1.55–3.13 μg/mL. However, the compound showed weak to no inhibitory activity against Gram-negative bacteria, such as *Pseudomonas aeruginosa* IFO3455 (MIC ≥ 100 μg/mL), *Proteus vulgaris* (MIC = 100 μg/mL), and *E. coli* K12 (MIC = 100 μg/mL) [30]. In the present study, we isolated LQM-A from a marine actinomycete *S. bacillaris* and confirmed the antibacterial activities of LQM-A against *S. aureus* ATCC25923 and *E. coli* ATCC25922, with MIC values of 0.06 μg/mL and 16 μg/mL, respectively, which were ten times higher than those reported previously. These differences may be attributed to distinctions among strains or the purity of the compound. The potential for the development of resistance to compounds **1**–**4** in the MRSA strain *S. aureus* ATCC43300 was also investigated. As shown in Table 2, ciprofloxacin showed potent antibacterial activities against MRSA strain *S. aureus* ATCC43300 (MIC = 0.25 μg/mL). However, based on multi-step (30-passage) resistance selection studies in the presence of ciprofloxacin, a steady increase in MIC for ciprofloxacin against *S. aureus* ATCC43300 during the passage experiment was observed (Figure 2a). Reportedly, topoisomerase IV is the primary target of fluoroquinolones in *S. aureus*, which may be due to the greater sensitivity of topoisomerase IV to these agents [31]. Furthermore, resistance from an altered DNA gyrase requires resistant topoisomerase IV for expression [31,32]. By contrast, during the passage experiment, MIC values for compounds **1**, **3**, and **4** showed no more than a 4-fold increase from the starting MIC value, indicating that no resistance was detected over the 30 passages.

LQM-A is known as an inhibitor of the serine/threonine kinase AKT, which binds to T-loop cysteines in cancer cells, but its mechanism has not been well characterized in bacteria [33]. The mechanism of LQM-A is generally assumed to be related to the bacterial cell membrane based on previous research [23,24]. However, LQM-A did not affect membrane potential in our membrane permeabilization assay (Figure 3b). Thus, the mode of action of LQM-A appears to be the induction of DNA damage or inhibition of protein synthesis due to its structural similarities with Dox and Tet. In the present study, we demonstrated that the mode of action of LQM-A approximates that of Dox using a dual-reporter system. LQM-A-induced expression of RFP (shown in green pseudo-color), indicating that it caused the SOS response to DNA damage (Figure 3a). This result indicated that the tetracyclic structure of LQM-A, containing three six-membered rings and one five-membered ring, is relatively flat, like that of Dox. The mechanism of inducing DNA damage was identified as intercalation into DNA using the fluorescence intercalator EtBr. However, the preference of DNA base pairs for intercalation was minor (Figure 4). The preference of LQM-A among DNA regions for intercalation should be studied further via docking simulation and DNase Ι footprinting assay to clearly characterize the correlation between DNA base pairs and LQM-A.

## 4. Materials and Methods

### 4.1. General Experimental Equipment

^1^H, ^13^C, and 2D (COSY, HSQC, HMBC) nuclear magnetic resonance (NMR) spectra were measured in methanol-*d*_4_ using Bruker Avance 600 MHz spectrometers (Bruker, Billerica, MA, USA) at the National Instrumentation Center for Environmental Management (NICEM) of Seoul National University. Liquid chromatography/mass spectrometry (LC/MS) data were obtained with an Agilent Technologies 6130 quadrupole mass spectrometer (Agilent Technologies, Santa Clara, CA, USA) coupled with an Agilent Technologies 1200 series high-performance liquid chromatography (HPLC) instrument. High-resolution fast atom bombardment (HR-FAB) mass spectra were recorded on a Jeol JMS-700 high-resolution mass spectrometer (Jeol, Tokyo, Japan) at the National Center for Inter-University Research Facilities (NCIRF) of Seoul National University. HPLC was performed using a Gilson 321 HPLC pump with a Gilson UV/VIS-151 detector (Gilson, Middleton, WI, USA). All solvents used were of spectroscopic grade or were distilled prior to use.

### 4.2. Taxonomic Identification of the Lactoquinomycin-Producing Microorganism

Marine sediment samples were collected from the shoreline of Jeju Island, Korea. The air-dried sediment (1 g) was mixed with 10 mL of sterilized artificial seawater and shaken at 150 rpm for 30 min at 25 °C. The suspension was serially diluted with sterilized artificial seawater and 0.1 mL volumes of three appropriate dilutions were spread onto actinomycete isolation agar (HIMEDIA, Mumbai, India) plates supplemented with 100 μg/mL cycloheximide, 50 μg/mL nalidixic acid (Sigma-Aldrich, St. Louis, MO, USA) and artificial seawater. Plates were incubated at 28 °C for 2 to 3 weeks. To obtain single strains, colonies were transferred several times on fresh agar plates by streaking.

The isolated bacterial strain MBTC38 was identified using standard molecular biological protocols, including DNA amplification and sequencing of the 16S rDNA region [34]. Briefly, genomic DNA was prepared from MBTC38 mycelium using the i-Genomic BYF DNA Extraction Mini Kit (Intron Biotechnology, Seoul, Korea) according to the manufacturer’s protocol. Polymerase chain reaction (PCR) amplification using the primers 27F (5′-AGAGTTTGATCCTGGCTCAG-3′) and 1429R (5′-GGTTACCTTGTTACGACTT-3′) was performed under the following conditions: pre-denaturation (95 °C, 5 min); 30 cycles of denaturation (95 °C, 15 s), annealing (50 °C, 30 s), and extension (72 °C, 1 min 30 s); and final extension (72 °C, 5 min). The nucleotide sequence was deposited in GenBank under accession number MK402083.1 and aligned according to the nucleotide basic logic alignment search tool (BLAST, version 2.10) of the National Center for Biotechnology Information (NCBI).

### 4.3. Cultivation, Extraction, and Isolation of Compounds

*Streptomyces bacillaris* strain MBTC38 was sporulated on colloidal chitin agar plates (4 g chitin, 0.7 g K_2_HPO_4_, 0.5 g MgSO_4_·7H_2_O, 0.3 g KH_2_PO_4_, 0.01 g FeSO_4_·7H_2_O, 0.001 g MnCl_2_·4H_2_O, 0.001 g ZnSO_4_·7H_2_O, 20 g agar, and 17 g sea salt in 1 L distilled water) at 28 °C for 10 days. Mature spores were inoculated into 500 mL colloidal chitin liquid medium and incubated at 28 °C for 7 days on a rotatory shaker. The entire culture (60 L) was filtered through filter paper and extracted with an equal volume of ethyl acetate twice. The organic solvents were evaporated to dryness under reduced pressure to obtain 1.9 g of total extracts. Based on the results of the antibacterial activity assay, the entire extract (1.9 g) was separated through reversed-phase HPLC (Agilent Eclipse XDB-C18, 5 μm, 9.4 × 250 mm) under gradient solvent conditions ranging from 20% aqueous methanol to 100% methanol with 0.1% trifluoroacetic acid (ultraviolet detection at 254 nm, flow rate: 2 mL/min). After separation of compounds **1**–**4**, each compound was further purified through reversed-phase HPLC under isocratic conditions to obtain compounds **1** (*t*R: 23 min, 8.4 mg), **2** (*t*R: 21 min, 3.8 mg), **3** (*t*R: 24 min, 11.6 mg), and **4** (*t*R: 25 min, 5.1 mg).

Lactoquinomycin A (**1**): ^1^H NMR (CDCl_3_) δ_H_ 12.27 (1H, br s), 7.91 (1H, d, *J* = 7.8 Hz), 7.73 (1H, d, *J* = 7.8 Hz), 5.25 (1H, d, *J* = 2.9 Hz), 5.07 (1H, q, *J* = 7.0 Hz), 4.90 (1H, br d, *J* = 9.6 Hz), 4.69 (1H, dd, *J* = 4.9, 3.0 Hz), 3.63 (1H, m), 3.54 (1H, m), 3.43 (dd, *J* = 9.1, 9.1 Hz), 3.14 (dd, *J* = 17.8, 5.2 Hz), 2.97 (1H, dd, *J* = 17.6, 5.1 Hz) 2.9 (3H, br s), 2.76 (3H, br s), 2.69 (1H, d, *J* = 17.6 Hz), 2.54 (1H, m), 1.56 (3H, d, *J* = 6.9 Hz), 1.47 (3H, d, *J* = 6.1 Hz), 1.29, (1H, m); ^13^C NMR (CDCl_3_) δ_C_ 188.5, 181.1, 173.9, 157.7, 149.6, 136.5, 135.5, 133.8, 130.5, 119.9, 114.3, 77.7, 71.1, 70.4, 68.5, 68.0, 66.4, 66.2, 40.1, 39.6, 36.9, 29.3, 18.6, 18.1; HRFABMS *m*/*z* 458.1811 [M + H]^+^ (calcd for C_24_H_28_NO_8_, 458.1811).

Lactoquinomycin B (**2**): ^1^H NMR (MeOH-*d*_4_) δ_H_ 7.97 (1H, d, *J* = 7.9 Hz), 7.68 (1H, d, *J* = 7.9 Hz), 5.48 (1H, d, *J* = 4.3 Hz), 5.01 (1H, q, *J* = 7.2 Hz), 4.71 (1H, m), 4.62 (1H, dd, *J* = 6.2, 4.2 Hz), 3.60 (1H, m), 3.45 (1H, m), 3.04 (1H, dd, *J* = 18.4, 6.2 Hz), 2.91 (3H, br s), 2.80 (3H, br s), 2.64 (1H, m), 2.52 (1H, ddd, *J* = 12.4, 3.8, 2.2 Hz), 2.43 (1H, d, *J* = 18.4 Hz), 1.67 (1H, m), 1.62 (1H, d, *J* = 6.8 Hz), 1.41 (1H, d, *J* = 6.1 Hz); ^13^C NMR (MeOH-*d*_4_) δ_C_ 195.2, 187.3, 175.8, 157.6, 136.6, 133.8, 130.8, 119.1, 114.0, 77.2, 71.0, 70.3, 69.8, 67.1, 65.0, 64.4, 64.3, 60.5, 40.7, 36.0, 34.9, 28.9, 16.8, 13.3; HRFABMS *m*/*z* 474.1761 [M + H]^+^ (calcd for C_24_H_28_NO_9_, 474.1759).

N-methyl lactoquinomycin A (**3**): ^1^H NMR (MeOH-*d*_4_) δ_H_ 7.93 (1H, d, *J* = 7.8 Hz), 7.73 (1H, d, *J* = 7.8 Hz), 5.34 (1H, d, *J* = 3.0 Hz), 5.08 (1H, q, *J* = 6.8 Hz), 5.02 (1H, br d, *J* = 10.5 Hz), 4.79 (1H, dd, *J* = 5.1, 2.7 Hz), 3.57 (1H, dq, *J* = 9.1, 6.1 Hz), 3.40 (1H, m), 3.28 (1H, m), 3.14 (1H, dd, *J* = 17.8, 5.1 Hz), 2.75 (3H, s), 2.65 (1H, ddd, *J* = 12.0, 4.2, 2.0 Hz), 2.54 (1H, d, *J* = 17.8 Hz), 1.58 (1H, d, *J* = 6.8 Hz), 1.42 (1H, d, *J* = 6.2 Hz); ^13^C NMR (MeOH-*d*_4_) δ_C_ 188.7, 181.5, 175.9, 157.5, 149.6, 136.5, 135.4, 133.2, 130.8, 118.7, 114.6, 77.1, 72.2, 71.0, 69.5, 66.8, 66.4, 60.7, 36.3, 32.6, 29.2, 17.2, 16.9; HRFABMS *m*/*z* 444.1645 [M + H]^+^ (calcd for C_23_H_26_NO_8_, 444.1653).

Menoxymycin A (**4**): ^1^H NMR (CDCl_3_) δ_H_ 12.21 (1H, br s), 7.89 (1H, d, *J* = 7.5 Hz), 7.70 (1H, d, *J* = 7.5 Hz), 5.27 (1H, d, *J* = 2.7 Hz), 5.06 (1H, q, *J* = 6.2 Hz), 4.91 (br d, *J* = 9.8 Hz), 4.70 (1H, dd, *J* = 4.7, 2.5 Hz), 4.12 (1H, ddd, *J* = 11.5, 8.3, 3.0 Hz), 3.93 (1H, dd, *J* = 9.1, 9.1 Hz), 3.67 (1H, dd, *J* = 8.8, 6.1 Hz), 3.63 (3H, s), 3.58 (3H, s), 2.98 (1H, dd, *J* = 16.2, 4.8 Hz), 2.76 (1H, ddd, *J* = 11.5, 3.0, 2.0), 2.70 (1H, d, *J* = 16.2 Hz), 1.54 (1H, d, *J* = 6.5 Hz), 1.48 (1H, m), 1.47 (1H, d, *J* = 5.6 Hz); ^13^C NMR (CDCl_3_) δ_C_ 188.4, 181.1, 174.0, 157.7, 149.6, 136.3, 135.5 133.8, 130.6, 119.8, 114.3, 78.5, 77.3, 71.2, 71.0, 68.6, 66.5, 66.2, 56.6, 55.5, 36.9, 32.2, 18.5, 17.9; HRFABMS *m*/*z* 474.1757 [M + H]^+^ (calcd for C_24_H_28_NO_9_, 474.1759).

### 4.4. Antibacterial Activity Assays

Antibacterial activity assays were carried out according to the methods of the Clinical and Laboratory Standard Institute (CLSI) [35]. Microorganisms obtained from the American Type Culture Collection (ATCC) and the stock Culture Collection of Antimicrobial Resistant Microorganisms (CCARM; clinical isolates; Seoul Women’s University, Seoul, Republic of Korea) were used for antibacterial activity assays: Gram-positive bacteria (*Staphylococcus aureus* ATCC25923, *Enterococcus faecalis* ATCC19433, and *E. faecium* ATCC19434), Gram-negative bacteria (*Salmonella enterica* ATCC14028, *Klebsiella pneumoniae* ATCC10031, and *Escherichia coli* ATCC25922), methicillin-sensitive *S. aureus* (MSSA; strains CCARM0027, 0204, 0205, and 3640), and methicillin-resistant *S. aureus* (MRSA; strains CCARM3089, 3090, 3634, 3635, and ATCC43300). Each bacterium was cultured overnight in cation-adjusted Mueller Hinton broth (MHBc; BD Difco, Sparks, MD, USA) at 37 °C, collected via centrifugation, diluted in MHBc, and adjusted to match the turbidity of a 0.5 McFarland standard at 625 nm wavelength [approximately 5 × 10^6^ colony forming units (cfu)/mL of the test bacterium]. Stock solutions of the compound were prepared in dimethyl sulfoxide (DMSO) at 12.8 mg/mL. Each stock solution was diluted in MHBc to concentrations ranging from 0.03 to 32 μg/mL. The final DMSO concentration was maintained at 1% by adding DMSO to the medium, according to CLSI guidelines. In each well of a 96-well plate, 90 μL of MHBc containing the test compound was mixed with 10 μL of broth containing the test bacterium (final concentration: 5 × 10^5^ cfu/mL). The plates were incubated for 24 h at 37 °C. The minimum inhibitory concentration (MIC) value was defined as the lowest concentration of test compound that prevented cell growth. Ampicillin (Duchefa, Amsterdam, Netherlands), tetracycline, daptomycin, vancomycin, platensimycin, linezolid, and ciprofloxacin (Sigma-Aldrich, St. Louis, MO, USA) were used as positive controls.

### 4.5. Multi-Step Resistance Development Assay

To identify drug susceptibility and the appearance of strains resistant to lactoquinomycin A and its analogs, multi-step resistance development experiments were carried out based on methods described previously [36] with minor modifications. Each test compound, including compounds **1**–**4** and ciprofloxacin (Cip; dissolved in DMSO at 12.8 mg/mL), was serially 2-fold diluted in MHBc based on previously determined MIC values. Overnight cultures of MRSA strain ATCC43300 were prepared at 106 cfu/mL in MHBc based on the 0.5 McFarland standard. Next, 50 μL of bacterial solution was inoculated into 50-μL dilution series of each compound in a 96-well plate (final concentration: 5 × 10^5^ cfu/mL). Following incubation at 37 °C for 24 h, the MIC was determined. The cultures were passaged daily for 30 days, using 50-μL inocula (10^6^ cfu/mL) from wells of the highest antibiotic concentration showing bacterial growth from the previous passage to inoculate a fresh series of dilutions.

### 4.6. Time-Kill Kinetics Assay

A time-kill kinetics assay of lactoquinomycin A was conducted according to the following procedures [37]. A single colony of overnight-cultured MRSA strain ATCC43300 was inoculated into fresh MHBc medium and incubated to the turbidity of the 0.5 McFarland standard. Additions of 8× the MICs of lactoquinomycin A (MIC = 0.25 μg/mL), tetracycline (MIC = 0.25 μg/mL), ciprofloxacin (MIC = 0.25 μg/mL), and vancomycin (MIC = 1 μg/mL) were made to MHBc containing 5 × 10^5^ cfu/mL of ATCC43300. These mixtures were incubated at 37 °C, and 200-μL aliquots of medium were collected at 0, 1, 2, 4, 8, and 24 h and inoculated on MHBc agar plates. The cfu values of the tested samples were determined following overnight incubation at 37 °C. The assay was conducted for three independent experiments and log_10_ (cfu/mL) was plotted against time.

### 4.7. Membrane Permeabilization Assay

Membrane potential was monitored using the fluorescent dye DiSC_3_(5) (3,3′-dipropylthiadicarbocyanine iodide) (Anaspec, Fremont, CA, USA) according to the following procedures [38]. *S. aureus* ATCC25923 was grown to mid-logarithmic phase (0.5 to 0.6 optical density at 600 nm) in MHBc at 37 °C. Cells were collected through centrifugation (5500× *g*, 10 min) and washed with buffer (5 mM HEPES, 5 mM glucose, pH 7.2). Following resuspension in the same buffer (optical density of 0.05 at 600 nm), DiSC_3_ (5) and KCl were added to final concentrations of 0.4 μM and 100 mM, respectively. The mixture was incubated until stabilization of fluorescence and treated with 4× the MICs of lactoquinomycin A (MIC = 0.06 μg/mL) and nisin (MIC = 4 μg/mL), using the latter as a positive control and DMSO as a negative control. Fluorescence was measured using a microplate reader FLx800 (BioTek, Winooski, VT, USA) at wavelengths of 622 nm for excitation and 670 nm for emission.

### 4.8. Detection of the Mode of Action with the Dual-Reporter System

To assess the mode of action of LQM-A, an in vivo dual-reporter assay designed to distinguish between compounds that inhibit translation and those that induce DNA damage was employed [21,22]. The plasmid, pDualrep2, was kindly provided by Prof. Ilta A. Osterman (Lomonosov Moscow State University, Moscow, Russia). To construct the dual-reporter system, pDualrep2 was transformed into *E. coli* DH5α with the heat-shock method (42 °C, 90 s) [39]. Tested antibiotic solutions at the MIC (doxorubicin = 4 μg/mL, erythromycin = 4 μg/mL, lactoquinomycin A = 4 μg/mL, and tetracycline = 0.25 μg/mL) were dropped onto an MHBc agar plate that already contained a lawn of the reporter strain. The agar plate was analyzed using the Chemi-Doc (Bio-Rad, Hercules, CA, USA) imaging system in Cy3 mode for red fluorescent protein (RFP) and Cy5 mode for Katushka2S after incubation overnight at 37 °C.

### 4.9. Ethidium Bromide (EtBr) Displacement Assay

To confirm whether lactoquinomycin A interacts with DNA, an EtBr displacement assay was performed according to previously reported procedures [25]. First, S. aureus strain ATCC25923 was selected, and its DNA was rapidly extracted using the Tris-EDTA-NaCl-Triton X100 (TENT) method [40]. The strain was cultured in 100 mL tryptic soy broth (TSB) at 37 °C for 16 h and then centrifuged (11,000× *g*, 10 min). The pellets were suspended in 1 mL TENT buffer containing 10 mM Tris-Cl, 0.1 M NaCl, 1 mM EDTA, and 5% [*v*/*v*] Triton X100 at pH 8.0. Then, the cell suspension was boiled at 100 °C for 5 min. Following centrifugation, the supernatant was mixed with cold 95% ethanol and stored at −20 °C for 20 min. The DNA pellet was dissolved in distilled water after the solution was re-centrifuged and air-dried. The DNA concentration was measured using a NanoDrop spectrophotometer (ACTGene, Piscataway, NJ, USA). We used 20-mers of poly-AT and poly-GC double-stranded DNA synthesized by Cosmogenetech (Seoul, Korea). Next, 70 μL of 8.57 μM EtBr solution containing 0.1 M Tris-Cl and 0.1 M NaCl at pH 8.0 (final EtBr concentration: 6 μM) was added to 96-well black plates, and 10 μL of 1 mg/mL prepared DNA solution was mixed with the EtBr solution. Then, 20 μL of Tris buffer containing up to 125 μM of lactoquinomycin A was added to each well and equilibrated at room temperature for 30 min. The intensity of fluorescence was read (excitation: 545 nm, emission: 595 nm) using a BioTek FLx800 microplate reader. The procedure was conducted in triplicate and a graph of relative fluorescence intensity against the concentrations of test compounds was plotted.

### 4.10. DNA Mobility Shift Assay

The rapid and sensitive DNA mobility shift assay was performed using previously described procedures [41]. The plasmid pUC 19 (2.7 kb) was used for double-strand DNA intercalation. Plasmid DNA was isolated using the DNA-maxi SV Plasmid DNA Purification Kit (Intron Biotechnology, Seoul, Korea), and the concentration was measured using a NanoDrop spectrophotometer. The assay was conducted using 1 μL (100 ng/μL) plasmid DNA mixed with various concentrations of lactoquinomycin A solution (from 0× to 8× MIC). The total volume of each reaction was brought to 10 μL with Tris buffer (0.1 M Tris-Cl, 0.1 M NaCl, pH 8.0). For time-dependent experiments, lactoquinomycin A was added at the MIC value and incubated for 0 to 120 min. To assess intercalation activity, the gel was stained with EtBr for 30 min after electrophoresis, using a 1% (w/v) agarose gel without EtBr and 0.5× Tris-Borate-EDTA (TBE) buffer at 50 V for 2 h. The gel was monitored using the Chemi-Doc imaging system with ultraviolet light.

## 5. Conclusions

A marine-derived actinomycete (*Streptomyces* sp. MBTC38) exhibiting antibacterial activities was investigated in the present study. The 16S rDNA sequence of the isolate indicated that it was most closely related to *Streptomyces bacillaris*. Furthermore, lactoquinomycin A (LQM-A) (**1**) and its derivatives (**2**–**4**) were isolated from the ethyl acetate extract. Their structures were determined using extensive spectroscopic methods, as well as comparisons with previously reported data. These compounds showed potent antibacterial activities against Gram-positive bacteria including MRSA. LQM-A exhibited the most significant inhibitory activity against MRSA, with MIC values of 0.25–0.5 μg/mL, which are more potent than values of major classes of antibiotics that include vancomycin and linezolid. During a passage experiment, the MIC value for LQM-A showed no more than a 4-fold increase from the starting MIC value, indicating that no resistance was detected over the 30 passages. The in vivo dual-reporter assay with LQM-A showed pseudo-green fluorescence similar to that of Dox, indicating that LQM-A induces DNA damage rather than inhibiting protein synthesis. The gel mobility shift assay showed that LQM-A switched plasmid DNA from the supercoiled form to the relaxed form. Our biochemical research supports LQM-A as a promising antibacterial drug and helps to elucidate the mode of action of its analogs. However main pharmacological focus remains in anticancer activity due to LQM-A showed cytotoxicity at low concentrations against human leukemia cell line K562 (IC_50_ = 33 ng/mL) and murine lymphoma cell line L5178Y (IC_50_ = 20 ng/mL) [30]. Therefore, modifications of the side chain can potentially be exploited for modulation of the biological activity of lactoquinomycin compounds. Lactoquinomycins derived from marine actinomycetes should be further investigated.

## Figures and Tables

**Figure 1 marinedrugs-19-00007-f001:**
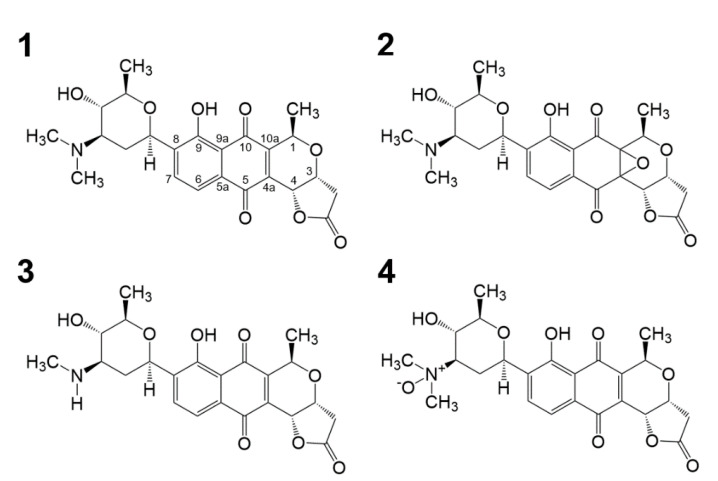
The structures of compounds **1**–**4**. Lactoquinomycin A (**1**), lactoquinomycin B (**2**), N-methyl lactoquinomycin A (**3**), and menoxymycin A (**4**).

**Figure 2 marinedrugs-19-00007-f002:**
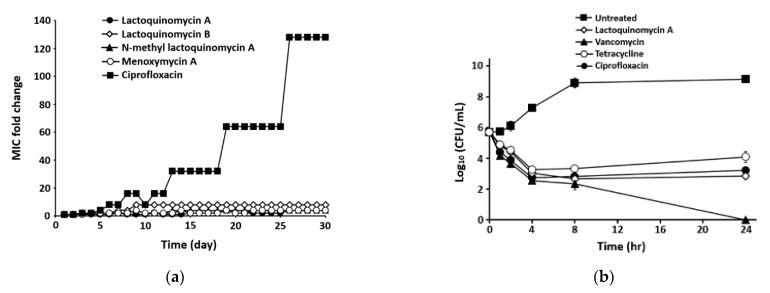
Antibacterial activities of lactoquinomycins (**1**–**4**) against MRSA strain ATCC43300. (**a**) The multi-step resistance development assay was conducted for 30 days with sub-MIC levels of four lactoquinomycins and ciprofloxacin. (**b**) The time-kill kinetics experiment was performed for 24 h with 8× MIC levels of LQM-A (MIC = 0.25 μg/mL), tetracycline (MIC = 0.25 μg/mL), ciprofloxacin (MIC = 0.25 μg/mL), and vancomycin (MIC = 1 μg/mL). Each point indicates the mean ± standard deviation (SD) of three independent experiments.

**Figure 3 marinedrugs-19-00007-f003:**
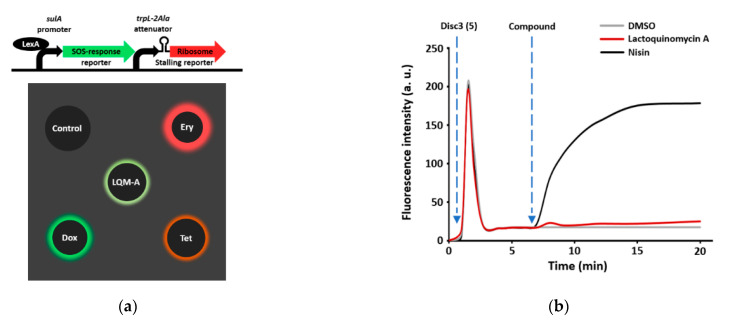
LQM-A induces DNA damage. (**a**) The dual-reporter system employed MIC levels of LQM-A, Ery, Tet, and Dox (4 μg/mL, 4 μg/mL, 4 μg/mL, and 0.25 μg/mL, respectively). The reporter strain was spread on an MHBc agar plate; then, MICs of each compound were dropped onto it and incubated overnight at 37 °C. The plate was analyzed with the Chemi-Doc imager in Cy3 mode for red fluorescent protein (RFP) and Cy5 mode for Katushka2S. (**b**) The membrane permeabilization assay was performed with 4× the MICs of LQM-A (MIC = 0.06 μg/mL) and nisin (MIC = 4 μg/mL), using the latter as a positive control and DMSO as a negative control against *S. aureus* ATCC25923. Fluorescence was applied at wavelengths of 622 nm for excitation and 670 nm for emission.

**Figure 4 marinedrugs-19-00007-f004:**
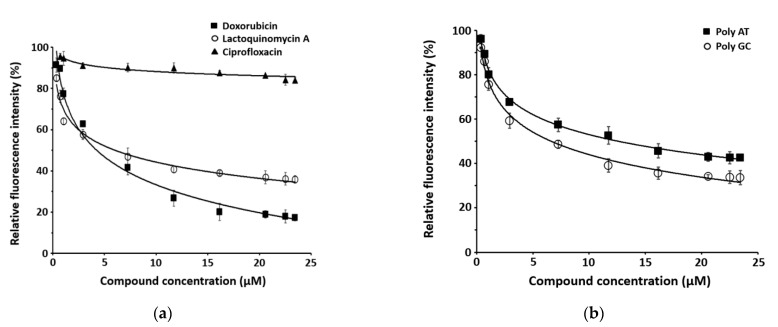
EtBr displacement assay based on fluorescence spectroscopy. (**a**) The fluorescence intensity of the EtBr and *S. aureus* ATCC25923 DNA complex was measured at wavelengths of 545 nm for excitation and 595 nm for emission with increasing concentrations of LQM-A. Cip and Dox were used as control antibiotics. (**b**) 20-mers of poly-AT and poly-GC DNA were applied to identify preference among DNA intercalation regions. All experiments were conducted independently in triplicate and each point represents the mean ± SD.

**Figure 5 marinedrugs-19-00007-f005:**
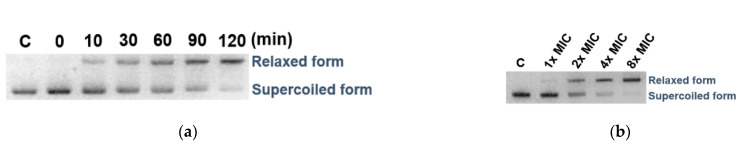
Gel mobility shift assay in the presence of LQM-A. The assay was performed with pUC 19 plasmid (**a**) at various time points (0, 10, 30, 60, 90, and 120 min) and (**b**) with different concentrations of LQM-A from 0× to 8× MIC. C: untreated pUC 19.

**Table 1 marinedrugs-19-00007-t001:** Results of antibacterial activity test.

Microorganism	MIC (μg/mL)					
	1	2	3	4	Amp ^a^	Tet ^a^
*Staphylococcus aureus* ATCC25923	0.06	2	0.5	1	0.06	0.25
*Enterococcus faecalis* ATCC19433	2	>32	2	4	0.5	0.5
*Enterococcus faecium* ATCC19434	2	32	2	4	0.5	0.5
*Salmonella enterica* ATCC14028	0.03	1	0.25	0.13	0.03	0.13
*Klebsiella pneumoniae* ATCC10031	>32	>32	32	>32	>32	0.25
*Escherichia coli* ATCC25922	16	>32	32	>32	8	0.5

MIC, minimum inhibitory concentration. ^a^ Positive control. Amp: ampicillin, Tet: tetracycline. DMSO (1%) was used as a negative control.

**Table 2 marinedrugs-19-00007-t002:** Antibacterial activities of compounds **1**–**4** against MSSA and MRSA strains.

Microorganism	MIC (μg/mL)								
	Dap	Van	Pla	Lin	Cip	1	2	3	4
CCARM0027 ^a^	4	0.5	4	2	0.25	0.25	2	1	1
CCARM0204 ^a^	1	0.5	4	1	0.25	0.13	1	0.25	0.5
CCARM0205 ^a^	0.5	0.5	2	1	0.25	0.06	1	0.25	0.5
CCARM3640 ^a^	8	0.25	4	2	0.25	0.13	2	1	1
CCARM3089 ^b^	>32	0.5	8	2	>32	0.25	4	1	2
CCARM3090 ^b^	32	0.06	8	1	>32	0.5	8	1	2
CCARM3634 ^b^	32	0.25	8	2	>32	0.25	4	1	1
CCARM3635 ^b^	>32	0.06	8	2	>32	0.25	4	0.5	0.5
ATCC43300 ^b^	>32	1	4	2	0.25	0.25	4	0.5	1

MIC, minimum inhibitory concentration. ^a^ Methicillin-sensitive *Staphylococcus aureus* (MSSA). ^b^ Methicillin-resistant *Staphylococcus aureus* (MRSA). Dap: daptomycin, Van: vancomycin, Pla: platensimycin, Lin: Linezolid, Cip: ciprofloxacin. DMSO (1%) was used as a negative control.

## Data Availability

The data presented in this study are available in the Appendix A.

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
