# Peer review of "Antibacterial Activity and Mode of Action of Lactoquinomycin A from Streptomyces bacillaris"

_marinedrugs, 2020, doi:10.3390/md19010007_

Round 1
Reviewer 1 Report
Review of the article: “Antibacterial activity and mode of action of lactoquinomycin A from Streptomyces bacillaris”
Submission ID marinedrugs-1042573
Drug resistance of bacterial pathogens is one of most important challenge of modern medicine. Thus, there is an urgent need to look for new antibiotics. The authors of the manuscript isolated a strain of Streptomyces bacillaris producing antibacterial metabolites - lactoquinomycin A and its’ derivatives. Moreover, they developed the method of purification of active substances, determined their antimicrobial potential against broad spectrum of pathogenic bacteria and performed some research aiming in explanation of modes of action of these metabolites. In my opinion the presented manuscript is interesting and well prepared. It can be accepted for publication in “Marine drugs”. However, I have several minor of importance critical remarks and comments. I would be grateful if the authors considered my comments preparing the final version of the manuscript. The detailed comments are presented below.
Detailed comments:
Abstract – Generally the abstract is well prepared. However, in my opinion some most important results should be presented in abstract – e.g. MIC values against Staphylococci. It also should be written that the agents exhibit activity against Gram-positive bacteria, activity against Gram-negative pathogens was low.
Introduction
Introduction is well prepared, no critical remarks
Materials and methods
Lines 226-238 – identification of producing microorganism – the authors presented the method of identification of producing strain. I would be grateful for some details about the story of selection of this strain – how was this strain isolated from natural reservoir “marine sediment”. Two – maximum three sentences would be required.
Line 240 – why the authors used this medium for growing of producing strain (was it possible to use another e.g. commercial available medium?)
Lines 260-262 – some more details about MSSA and MRSA strains would be required – are these clinical isolates? Except of one ATTC strain of course.
Line 290 – very high activity of all agents (8xMIC) were used for Time-Kill Kinetic assay. Please explain.
Line 330 – what was the aim of using of poly – AT and poly – GC for the study - it should be clearly written.
Results – generally the results are interesting and well presented. Only few comments.
Lines 67-77 – it should be clearly written that these are previously known substances.
Figure 2 (b) – I do not know if it was necessary to use three other known agents for this assay as controls.
Figure 4 (a) – there are only two symbols in the legend and in the picture there are three curves.
Discussion – in my opinion the current version of discussion is rather week. The authors repeated some results. The authors should present comparison of their results with results obtained by other authors (at least short discussion should be presented). Moreover some perspectives and limitations of application of “new metabolites” in clinical scenario should be presented.
Final decision – minor revision.
Author Response
Thank you very much for your careful and valuable review of our manuscript. We made all revisions and corrections as far as we could. I hope this is the right answer for your request. What follows is our response to reviewer#1’s critique with the explanation of the changes implemented in the paper and a rebuttal when appropriate.
Comment 1:
Abstract – Generally the abstract is well prepared. However, in my opinion some most important results should be presented in abstract–e.g. MIC values against Staphylococci. It also should be written that the agents exhibit activity against Gram-positive bacteria, activity against Gram-negative pathogens was low.
Answer)
We appreciate Reviewer’s kind comments. According to the reviewer’s comments, the MIC values and activities of lactoquinomycins against Gram-positive and Gram-negative bacteria are described in the revised version (lines 16-21).
Comment 2:
Materials and methods
Lines 226-238 – identification of producing microorganism – the authors presented the method of identification of producing strain. I would be grateful for some details about the story of selection of this strain – how was this strain isolated from natural reservoir “marine sediment”. Two – maximum three sentences would be required.
Answer)
The history of selection of the lactoquinomycin-producing microorganism, Streptomyces bacillaris strain MBTC38, from marine sediments is described in the revised version (lines 244-251).
Comment 3:
Line 240 – why the authors used this medium for growing of producing strain (was it possible to use another e.g. commercial available medium?)
Answer)
The genus Streptomyces has been reported to produce different metabolites depending on composition of medium. We tested eight media for choosing the most efficient medium in producing antimicrobial metabolites. Colloidal chitin media was the best and used in this study.
Comment 4:
Lines 260-262 – some more details about MSSA and MRSA strains would be required – are these clinical isolates? Except of one ATTC strain of course.
Answer)
MSSA and MRSA strains obtained from the stock Culture Collection of Antimicrobial Resistant Microorganisms (CCARM; Seoul Women’s University, Seoul, Republic of Korea) were clinical isolates. Thus, we described isolated source at line 309.
Comment 5:
Line 290 – very high activity of all agents (8xMIC) were used for Time-Kill Kinetic assay. Please explain.
Answer)
8x MIC was used for time-kill kinetics assay in many papers including reference 35. We also performed time-kill kinetics assay with 4x MIC and 8x MIC of all agents and overall tendency of inhibition was similar. Thus, in this paper, we presented results using 8x MIC because the difference of four agents was clear in assay with 8x MIC.
Comment 6:
Line 330 – what was the aim of using of poly – AT and poly – GC for the study - it should be clearly written.
Answer)
We confirmed that LQM-A intercalates to genomic DNA of S. aureus (Figure 4a). Additionally, we wanted to ascertain DNA region (AT rich or GC rich or half and half) where LQM-A intercalates. Therefore, we performed EtBr displacement assay using poly-AT and poly-GC. In case of doxorubicin similar to LQM-A, poly-AT and poly-GC were used for confirming DNA intercalating region of doxorubicin. We presented this information at lines 156-159.
Comment 7:
Results
Lines 67-77– it should be clearly written that these are previously known substances.
Answer)
We appreciate Reviewer’s kind comments. To clear these problems, we summarized NMR, IR and MS data for known compounds 1–4 in the Results section (lines 84-85) and Materials and methods section (lines 277-304), and added supplementary materials in the revised version.
Supplementary Materials: Table S1: 13C NMR compare 1 with lactoquinomycin A in CDCl3., Table S2: 13C NMR compare 2 with lactoquinomycin B in MeOH-d4, CDCl3, respectively., Table S3: 13C NMR compare 3 with N-methyl lactoquinomycin A in MeOH-d4., Table S4: 13C NMR compare 4 with menoxymycin A in MeOH-d4., Figure S1: 1H NMR of compound 1 in CDCl3, Figure S2: 13C NMR of compound 1 in CDCl3, Figure S3: 1H NMR of compound 2 in MeOH-d4, Figure S4: 13C NMR of compound 2 in MeOH-d4, Figure S5: 1H NMR of compound 3 in MeOH-d4, Figure S6: 13C NMR of compound 3 in MeOH-d4, Figure S7: 1H NMR of compound 4 in CDCl3, Figure S8: 13C NMR of compound 4 in CDCl3.
Comment 8:
Figure 2 (b) – I do not know if it was necessary to use three other known agents for this assay as controls.
Answer)
We used three other known agents because we wanted to compare time-dependent inhibitory activities of LQM-A to others. Based on these results, we confirmed LQM-A inhibited MRSA strain ATCC 43300 more superior than tetracycline and ciprofloxacin up to 24 h.
Comment 9:
Figure 4 (a) – there are only two symbols in the legend and in the picture there are three curves.
Answer)
We revised the missing legend.
Comment 10:
Discussion – in my opinion the current version of discussion is rather week. The authors repeated some results. The authors should present comparison of their results with results obtained by other authors (at least short discussion should be presented). Moreover some perspectives and limitations of application of “new metabolites” in clinical scenario should be presented.
Answer)
According to the reviewer’s comments, we deleted some repeated information and added comparison between our results and previous data from other authors in second paragraph of ‘Discussion’ (lines 194-214). The perspectives and limitation of application of LQM-A was described in Conclusion (lines 419-424).
Reviewer 2 Report
In abstract it must be transferred to the aim of study. In Introduction some information about Streptomyces bacillaris missing. In the line 227 Material and methods must be bacterial strain Streptomyces bacillaris MBTC38. Also the same mistakes must be improved in all text.
In the Antibacterial assay is negative control missing. Why does the author use Ethidium bromide, this chemical is carcinogenic?
Why did the authors describe Streptomyces bacillaris strain MBTC38 when the GenBank accession number is MK402083.1?
Results are described very well.
Discussion must be improved. I miss some information about antibacterial activity of lactoquinomycins and antimicrobial resistance of antibiotics.
Conclusion must be as chapter 5 and need more detailed information.
Author Response
Thank you very much for your careful and valuable review of our manuscript. We made all revisions and corrections as far as we could. I hope this is the right answer for your request. What follows is our response to reviewer#2’s critique with the explanation of the changes implemented in the paper and a rebuttal when appropriate.
Comment 1:
In abstract it must be transferred to the aim of study. In Introduction some information about Streptomyces bacillaris missing. In the line 227 Material and methods must be bacterial strain Streptomyces bacillaris MBTC38. Also the same mistakes must be improved in all text.
Answer)
We appreciate Reviewer’s kind comments. According to the reviewer’s comments, we described the aim of study in abstract (lines 12-14). The informations about Streptomyces bacillaris are also given in Introduction (lines 56-64). We revised bacterial strain as a Streptomyces bacillaris strain MBTC38 in the text except for part of identification of MBTC38 strain.
Comment 2:
In the Antibacterial assay is negative control missing. Why does the author use Ethidium bromide, this chemical is carcinogenic?
Answer)
DMSO (1%) was used as a negative control and we described at legends of Tables 1-2. Ethidium bromide (EtBr) is carcinogenic. However, EtBr was reported to intercalate to DNA and utilized as a fluorescence dye for a fluorescent intercalator displacement assay in vitro. Therefore we used EtBr as a fluorescence dye whether LQM-A intercalates to DNA.
Comment 3:
Why did the authors describe Streptomyces bacillaris strain MBTC38 when the GenBank accession number is MK402083.1?
Answer)
We isolated MBTC38 strain from the marine sediment, Jeju island, Republic of Korea. 16S rDNA sequencing was performed to identify MBTC38 strain. Based on the NCBI 16S rDNA blast, MBTC38 strain showed 100% identity to Streptomyces bacillaris strain NBRC13487 (type strain, GenBank accession number: NR041146.1) and ATCC15855 (type strain, GenBank accession number: CP029378.1). Although these strains showed 100% identity in 16S rDNA levels, at present, we can’t confirm these are same organisms or not and should be further investigated. Thus, we registered our strain as Streptomyces bacillaris strain MBTC38 (Genbank accession number, MK402083.1).
Comment 4:
Discussion must be improved. I miss some information about antibacterial activity of lactoquinomycins and antimicrobial resistance of antibiotics.
Answer)
According to the reviewer’s comments, we improved discussion and presented information about antibacterial activity of lactoquinomycins and incidence of resistance in the revised version (lines 194-214).
Comment 5:
Conclusion must be as chapter 5 and need more detailed information.
Answer)
According to the reviewer’s comments, we made ‘5. Conclusion’ section and added detailed information such as perspectives and limitation of LQM-A in clinical field (lines 405-424).
Round 2
Reviewer 2 Report
The authors accepted all comments. Thank you for great work.